# Cooperative Unmanned Aerial System Reconnaissance in a Complex Urban Environment and Uneven Terrain

**DOI:** 10.3390/s19173754

**Published:** 2019-08-30

**Authors:** Petr Stodola, Jan Drozd, Jan Mazal, Jan Hodický, Dalibor Procházka

**Affiliations:** 1Department of Intelligence Support, University of Defence, 66210 Brno, Czech Republic; 2Department of Tactics, University of Defence, 66210 Brno, Czech Republic; 3NATO Modelling & Simulation Centre of Excellence, Doctrine, Education & Training Branch, 00143 Roma, Italy; 4Department of Aircraft Technology, University of Defence, 66210 Brno, Czech Republic; 5Centre for Security and Military Strategic Studies, University of Defence, 66210 Brno, Czech Republic

**Keywords:** cooperative aerial reconnaissance, unmanned aerial systems, reconnaissance operation, simulated annealing, art gallery problem, waypoint optimization, occlusion effect

## Abstract

Using unmanned robotic systems in military operations such as reconnaissance or surveillance, as well as in many civil applications, is common practice. In this article, the problem of monitoring the specified area of interest by a fleet of unmanned aerial systems is examined. The monitoring is planned via the Cooperative Aerial Model, which deploys a number of waypoints in the area; these waypoints are visited successively by unmanned systems. The original model proposed in the past assumed that the area to be explored is perfectly flat. A new formulation of this model is introduced in this article so that the model can be used in a complex environment with uneven terrain and/or with many obstacles, which may occlude some parts of the area of interest. The optimization algorithm based on the simulated annealing principles is proposed for positioning of waypoints to cover as large an area as possible. A set of scenarios has been designed to verify and evaluate the proposed approach. The key experiments are aimed at finding the minimum number of waypoints needed to explore at least the minimum requested portion of the area. Furthermore, the results are compared to the algorithm based on the lawnmower pattern.

## 1. Introduction

Using unmanned aerial systems (UASs) for various information collection tasks has become a trend not only in the military but also in the civilian domain. In these types of tasks, UASs are deployed in some environment in order to explore or monitor a specified ground area of interest via their sensor systems, for example, video cameras. Fast technological progress is connected not only to hardware development (e.g., sensors) but also to software development. The former has the effect of higher fidelity and accuracy of collected information; the latter allows planning the task optimally using a fleet of multiple UASs that cooperate with one another to increase the quality of operational performance of the task at hand.

In the military, modern technologies, including unmanned systems, have been used for decades to increase situational awareness in the area of operations. The typical and frequent tasks, in which UASs are used, cover autonomous monitoring, exploration, reconnaissance and surveillance of a specified area of interest. Command, Control, Communication, Computer, Intelligence, Surveillance and Reconnaissance (C4ISR) systems are used for effective planning and the execution of these tasks.

### 1.1. Motivation

At the University of Defence, Czech Republic, the Tactical Decision Support System (TDSS) is being developed for the Czech Army as a part of the C4ISR system to plan military operations, including reconnaissance or surveillance operations using multiple unmanned aerial or ground vehicles [1].

The primary goal of the TDSS is to support commanders in their decision-making by providing the possible variants to fulfil their missions [2]. These variants are created and evaluated based on the models of military tactics implemented in the TDSS. A commander is supposed to use this system, if there is a model compatible with his/her mission, to plan and execute this mission in the optimal or near optimal manner; this depends on the selected optimization criterion. This topic is covered in detail in literature [3,4,5,6,7,8].

The success of a military operation may depend on the fast, accurate and complete reconnaissance of the area of interest on the battlefield. The contemporary military challenges are determined by the principles of the asymmetric warfare where belligerents, whose relative military power differs significantly, are engaged in combat that is often provoked by the belligerent with the smaller power in a specific environment such as high density urban areas or uneven mountainous terrain. The findings and results presented in this article extend the model for UAS reconnaissance implemented in the TDSS in order to increase its efficiency in a complex urban environment or uneven terrain.

### 1.2. Contribution and Organization of the Article

The model for planning and executing the reconnaissance operations in the TDSS is called Cooperative Aerial Reconnaissance (CAR). This model was proposed and developed by the authors of this article in the past. The objective of the model is to plan the routes of individual heterogeneous UASs in the fleet that are at the disposal of the commander to explore the complete area of interest in the optimal manner (which is as fast as possible in most cases), that is, every point lying inside the area of interest during the operation is covered by at least one of the sensors of the unmanned systems in the fleet.

The original CAR model assumes that the area of interest is perfectly flat. The problem of planning the routes by the CAR model is then transformed to the deployment of a finite number of waypoints. By visiting all the waypoints by at least one of the UASs during the operation, no space in the area of interest remains unexplored. However, the assumption of the flat area is rarely true in a real environment. Very uneven terrain, high buildings or other obstacles can occlude some space in the area of interest, which remains unexplored at the end of a reconnaissance operation.

In this article, the original CAR model is improved substantially to follow the current trend of asymmetric warfare and the resulting typical combat environment. Terrain and obstacles, which may reduce the space covered by the sensor systems of the UASs, are taken into account. The article is organized as follows–Section 2 reviews the literature and research connected with this topic. In Section 3, the original CAR model is introduced, the new enriched formulation of the problem is presented and the novel optimization method to deploy the waypoints is proposed. Section 4 sets out the experiments to compare the original and the new approach and to verify the proposed solution. Section 5 concludes the article.

## 2. Literature Review

Using the unmanned systems for information collection problems has become a research area with a broad range of scientific publications. The UASs are used in applications such as monitoring and inspection [9,10,11], surveying [12], mapping [13,14], communication and networking [15,16,17,18], traffic monitoring [19,20], construction [21,22] and many others.

A special class of information collection problems is monitoring or surveillance of a specific ground area by sensor systems of available UASs. Effective path planning is critical in these applications. A broad range of methods for route planning of UASs in applications such as monitoring, reconnaissance, persistent surveillance or searching for a target object have been proposed. An extensive survey of methods based on coverage path planning is provided in Reference [23]. The authors of Reference [24] provide an overview of publications on computational-intelligence-based methods used for UAS path planning. An extensive survey of literature connected with UASs routing and trajectory optimization is examined in Reference [25]; the authors identified 20 attributes that are common to the UAS path planning problems such as the type of fleet, mission characteristics and flight dynamics. Several methods for path optimization of UASs such as linear programming, dynamic programming, genetic algorithms and neural networks are evaluated in Reference [26] and relate to the aerial surveillance problem.

Cooperation of multiple UASs, when conducting the operation, can increase the performance and effectiveness in applications such as reconnaissance or surveillance significantly. The authors of Reference [27] proposed a swarm coordination bio-inspired algorithm and used it to search for a target object via a swarm of UASs equipped with imperfect sensors. A similar task was examined in Reference [28] where the authors proposed a distributed algorithm for searching moving targets through a fleet of cooperative UASs. In Reference [29], a UAS is modelled as an agent in terms of multi-agent system principle in order to provide a simulation environment that can be used to evaluate and analyse swarm control mechanisms. A distributed deep learning algorithm for real-time object identification and tracking used by cooperative UASs in a complex and adversarial environment involving motion, crowded scenes and varied camera angles is proposed in Reference [30].

Monitoring a specific ground area of interest represented by a polygon through sensors from UASs deployed in the area can be seen as a case of the Art Gallery Problem. This is a well-known NP-hard problem [31] that is studied extensively especially in two-dimensional space [32,33]. The three-dimensional case of the Art Gallery Problem is examined for example, in Reference [34]; the authors of this publication deal with determining the number of observers deployed on the ground to cover an area in the real terrain. In Reference [35], the authors consider the observers in the 3D Art Gallery Problem to be aerial drones that are placed above the uneven terrain in a certain range of altitudes; their approach to estimate the minimal number of sensors necessary and to calculate their locations is proposed using the 3-colouring method introduced in Reference [36]. 

The issue of monitoring and surveillance using UASs in a complex environment where obstacles (e.g., buildings) may occlude some space in the area of interest to be explored is studied by some publications. The authors of Reference [37] examine a problem of an UAS path planning in order to maximize the coverage of an urban area to discover and track multiple moving ground targets. The similar problem is examined in Reference [38] where authors consider tracking of a moving ground target in dense obstacle areas using UASs. The problem of an UAS occlusion-aware persistent surveillance in complex urban areas is considered for example, in References [39,40,41]. In Reference [42], the issue of real-world surveillance of a complex 3D environment to identify objects of interest via both ground and aerial autonomous robotic systems is considered; the solution connects the 3D Art Gallery Problem (area monitoring) and the Travelling Salesman Problem (path planning).

## 3. Problem Formulation and Solution

First, the original Cooperative Aerial Reconnaissance (CAR) model proposed by the authors in the past is revised in Section 3.1. Then, the new problem formulation considering a complex environment to be explored is stated in Section 3.2. Finally, the novel optimization algorithm that reflects the new formulation is proposed in Section 3.3.

### 3.1. Cooperative Aerial Reconnaissance

The first version of the CAR model was formulated in Reference [43]. The objective of the model is planning the reconnaissance operation, which is to explore a polygonal area of interest using a fleet of available UASs. The basic principles are as follows:A number of waypoints are deployed in the area of interest. Monitoring is conducted from these waypoints; each waypoint is visited by one of the UASs in the fleet that is suspended in the air (in the monitoring phase) at a certain height above the ground. The ground facing sensor of the UAS is able to monitor some circular area on the ground. The principles are similar to the well-known 2D Art Gallery Problem.The UASs are moving between waypoints along their trajectories so that all waypoints are visited by at least one of the UASs. The objective is to plan the trajectories of individual UASs so that the reconnaissance operation is conducted optimally. The optimality depends on the selected optimization criterion, which is mostly to execute the operation as fast as possible. All UASs have to return back to their initial positions after visiting all the waypoints on their routes. The principles are similar to the famous Min-Max Multi-Depot Vehicle Routing Problem [44].

In the first version, waypoints were deployed in the area of interest using a simple algorithm, which was fast but did not ensure the whole area to be covered completely. The improvement of the CAR model published in Reference [45] presents an algorithm, which reduces a number of waypoints and optimizes their locations so that the complete area of interest may be covered by at least one of the sensors. Another improvement [46] extended the model through smoothing the trajectories of unmanned systems in order to be able to use the model for Dubins vehicles that cannot change direction abruptly.

The approach mentioned above is shown in Figure 1. On the left (Figure 1a), the principle of monitoring the area from a waypoint by an UAS is shown. The yellow colour represents the monitored circular portion of the area with radius (R), which depends on the height of the UAS above the ground ($H_{AGL}$) and the angular field of view of the sensor ($A_{FOV}$). On the right (Figure 1b), an example situation is shown. The area of interest is bounded by the green colour, two UASs are available for the reconnaissance operation (violet dots) and 31 waypoints are deployed (green dots). Yellow circles represent the sensor ranges; it is clear that there is no space unexplored. The flight height of both UASs does not change when moving between waypoints along their paths (violet lines).

### 3.2. Problem Formulation

In this section, a new formulation of the CAR model, which assumes that the area of interest is not perfectly flat and contains obstacles, is stated.

Let $A_{I}\subset A$ be the area of interest to be explored as a part of the area of operations. The area of interest is defined by a polygon with or without holes. Let $U=\left\{U_{1},U_{2},\ldots ,U_{M}\right\}$ be a set of available UASs deployed in the area of operations where M is their number. Initial locations of each UAS are known. UASs in the fleet may differ by their flight parameters (e.g., average velocity) but must be equipped with sensors of the same quality and parameters. Let $W=\left\{W_{1},W_{2},\ldots ,W_{N}\right\}$ be a finite set of waypoints deployed in the area of operations, from which the area of interest is explored. Number and locations of waypoints are not known at the beginning.

Let $E=\left\{E_{1},E_{2},\ldots \right\}$ be an infinite set of elevations determined by the terrain for every point lying inside the area of operations. UASs move above the terrain at a constant height $H_{AGL}$, that is, their altitude at some location $E_{j}\in E$ is expressed by formula (1). The height $H_{AGL}$ must be-(a) bigger than the minimal height requested by a commander due to some tactical reasons (security, exposure, etc.) and (b) smaller than the maximal height limited by sensors used (e.g., because of their resolution and the following ability to discover and identify target objects).
(1)$$H_{i}=E_{j}+H_{AGL}\;\;\;\;\;\mathrm{for}\;\mathrm{all}\;U_{i}\in U\;\mathrm{and}\;E_{j}\in E$$

Let $\;O=\left\{O_{1},O_{2},\ldots ,O_{K}\right\}$ be a set of not-transparent objects (obstacles) lying inside the area of interest where K is their number; each object is represented by a polygon with or without holes (ground plan) and its height above the ground.

The maximum range (i.e., the radius of the projected circle area) R of the sensor covering the ground area is determined according to formula (2). It depends on height $H_{AGL}$ of an UAS above the ground level and the angular field of view $A_{FOV}$ of its sensor.
(2)$$R=H_{AGL}\cdot \tan\left(\frac{A_{FOV}}{2}\right)$$

However, uneven terrain and/or obstacles may cause that some parts in this range are occluded. The principle is shown in Figure 2 in a plane. The green colour represents the part of the terrain covered by the sensor; the red area is the part, which is occluded.

From each waypoint $W_{i}\in W$, some portion of the area of interest $A_{i}\subseteq A_{I}$ is monitored by an UAS $U_{j}\in U$ provided that this UAS flies through this waypoint. Every point $P\in A_{i}$ lying on the ground inside the area of interest and within the range of the sensor, which has a visual line of sight (VLOS) between the sensor and this point, is marked as visible; otherwise, it is marked as occluded. Formula (3) determines the total coverage of the area of interest obtained during the reconnaissance operation.
(3)$$C=\overset{N}{\underset{i=1}{\cup }}A_{i}\;\;\;\;\;\mathrm{for}\;\mathrm{all}\;W_{i}\in W$$

The number and locations of waypoints are not known at the beginning of the operation. Thus, the first optimization problem lies in determining the number N and locations for each waypoint $W_{i}\in W$ and is expressed by formulae (4) and (5) respectively. Formula (4) states that the number N should be as small as possible, whereas formula (5) states that the total coverage of the area of interest should be as large as possible. The former optimization criterion follows from the objective to conduct the operation as fast as possible. These two optimization criteria go against one another. Therefore, constraint (6) is employed, which states that the total coverage must meet the requirement of the commander for minimal coverage $C_{min}$. Then, the optimization problem can be formulated as follows—the objective is to find the minimum number of sensors that cover at least $C_{min}$ portion of the area of interest in case that the locations of the waypoints are optimized to cover as large an area as possible.
(4)Minimize(N)
(5)Maximize(|C|)
(6)$$\left|C\right|\geq C_{min}$$

The location of each waypoint is determined by two independent continuous variables $x_{i}$ and $y_{i}$. The domain of each variable depends on the area of interest—each waypoint $W_{i}\in W$ can be placed to an arbitrary point laying inside the circumscribed rectangle of the polygon defining the area of interest $A_{I}$. The configuration space of the optimization problem of N waypoints locations is given by 2N independent continuous variables: $S=\left(x_{1},y_{1},x_{2},y_{2},\ldots x_{N},y_{N}\right)$ where $x_{i}=\langle A_{I}{}_{minx},A_{I}{}_{maxx}\rangle$ and $y_{i}=\langle A_{I}{}_{miny},A_{I}{}_{maxy}\rangle$ for each waypoints $W_{i}\in W$.

When the number of waypoints and their locations are determined, the second optimization problem appears. Each waypoint $W_{i}\in W$ must be visited by exactly one of the UASs $U_{j}\in U$. Each UAS starts from its initial position, visits a number of waypoints and then returns back to its initial position. The objective is to find out, which waypoints will be visited by which UAS and in which order. Various optimization criteria, which depend on the commander, can be used—for example, minimization of the total time of the operation or minimization of the total distance travelled by all UASs. The former is expressed by formula (7), the latter by formula (8). The optimization problem is similar to the well-known NP-hard Multi-Depot Vehicle Routing Problem (MDVRP). An example situation of routes is shown in Figure 1b where two UASs are available and 31 waypoints are deployed.
(7)$$Minimize\left(\max\left(T_{1},T_{2},\ldots ,T_{M}\right)\right)$$
(8)$$Minimize\left(\overset{M}{\underset{i=1}{\sum }}L_{i}\right)$$
where $T_{i}$ is the time to fly along the route of UAS $U_{i}\in U$, $L_{i}$ is the length of the route of UAS $U_{i}\in U$, M is the number of UASs in the fleet.

### 3.3. Problem Solution

In this section, the algorithm to solve the first optimization problem formulated in the previous section is proposed. The second optimization problem is not dealt with in this article as the formulation has not changed compared to the original CAR model; the details of the metaheuristic algorithm proposed for solution can be found in Reference [44].

#### 3.3.1. Optimization of Waypoint Locations

First, the algorithm to optimize the locations of a certain number of waypoints to maximize the monitored area of interest is proposed—see formula (5) as an optimization criterion. The algorithm is based on the simulated annealing principle, which is a generic probabilistic method. It has proven to be successful in similar position optimization problems; see the publications of the authors [47,48].

The algorithm in pseudocode is shown in Figure 3. A location of each waypoint $W_{i}\in W$ is determined by two variables $x_{i}$ and $y_{i}$. Solution S is a vector of 2N independent variables:
$S=\left(x_{1},y_{1},x_{2},y_{2},\ldots x_{N},y_{N}\right)$. At the beginning of the algorithm (see point 1), a random solution S is generated, that is, each variable is randomly placed within the interval given by its domain. The algorithm works with the key parameter—temperature. In the main cycle of the algorithm (see points 3 to 13), the temperature is gradually dropped down (point 13). The termination of the algorithm is triggered when the temperature drops below the minimal temperature $T_{min}$. A number of steps are conducted within the main loop (see points 5 to 12) where the transformation of a current solution S and its replacement by a new solution $S^{\prime }$ is performed.

The evaluation of any solution S (see points 1 and 6) means the calculation of the coverage of the area of interest based on the actual locations of waypoints. The principle used in the current implementation is as follows. The area of interest is rasterized using a constant rasterization step. Each square in thus created grid is marked as visible when it lays inside the area of interest and, at the same time, there is a VLOS to the centre of this square from any waypoint $W_{i}\in W$, or it is marked as occluded otherwise. The sum of all squares marked as visible represents the coverage |C|.

The key principle of the algorithm is based on the transformation (see point 6) of solution S to solution $S^{\prime }=\left(x_{1}^{\prime },y_{1}^{\prime },x_{2}^{\prime },y_{2}^{\prime },\ldots x_{N}^{\prime },x_{N}^{\prime }\right)$ in successive steps. The transformation in each step lies in changing values of two variables $x_{t}$ and $y_{t}$ corresponding to a randomly selected waypoint $W_{t}\in W$ according to formula (9).
(9)$$x_{t}^{\prime }=\begin{array}{cc} \{\;\begin{array}{l} x_{t}+\mathrm{RandN}\left(\mu ,\sigma_{x}^{2}\right)\\ x_{t} \end{array} & \begin{array}{l} \mathrm{for}\;\mathrm{variable}\;W_{t}\in W\;\mathrm{selected}\;\mathrm{for}\;\mathrm{transformation}\\ \mathrm{otherwise} \end{array} \end{array}\;y_{t}^{\prime }=\begin{array}{cc} \{\;\begin{array}{l} y_{t}+\mathrm{RandN}\left(\mu ,\sigma_{y}^{2}\right)\\ y_{t} \end{array} & \begin{array}{l} \mathrm{for}\;\mathrm{variable}\;W_{t}\in W\;\mathrm{selected}\;\mathrm{for}\;\mathrm{transformation}\\ \mathrm{otherwise} \end{array} \end{array}$$

The changes are carried out by a random number generator with the Gaussian distribution; RandN in formula (9) is a random number generator with a normal distribution $N\left(\mu ,\sigma^{2}\right)$ where the mean μ=0 and the standard deviations $\sigma_{x}$ and $\sigma_{y}$ are set according to formula (10). The change depends on the current temperature T which lays in the interval $\left\langle T_{min},T_{max}\right\rangle$—the higher the temperature, the bigger the change. This principle ensures moving the variable in its full domain at the beginning phases of optimization, thus allowing to place the transformed solution $S^{\prime }$ within the whole configuration space. As the temperature is dropping down, the changes are smaller and smaller and the solution is approaching the optimal (or some local optimal) solution. In the final phases, the changes are small in the surroundings of the variable to tune the solution.
(10)$$\sigma_{x}=\frac{\left(T-T_{min}\right)\cdot \left(A_{I}{}_{maxx}-A_{I}{}_{minx}\right)}{T_{max}-T_{min}}\;\sigma_{y}=\frac{\left(T-T_{min}\right)\cdot \left(A_{I}{}_{maxy}-A_{I}{}_{miny}\right)}{T_{max}-T_{min}}$$
where T is the current temperature in interval $\left\langle T_{min},T_{max}\right\rangle$ and $A_{I}{}_{minx}$, $A_{I}{}_{miny}$, $A_{I}{}_{maxx}$, $A_{I}{}_{maxy}$ are the coordinates of the top-left corner and bottom-right respectively, of the circumscribed rectangle of the area of interest.

When a new solution $S^{\prime }$ is created and evaluated, this new solution is replaced (see point 9) by the old solution S with the probability based on the Metropolis criterion (see point 8) in formula (11). When this new solution is better than the old one, it is always replaced. Otherwise, the replacement is performed with probability based on the difference between the qualities of these solutions and the current temperature. This principle prevents the solution to be stuck in a local optimum.
(11)$$p\left(S\to S^{\prime }\right)=\begin{array}{cc} \{\;\begin{array}{l} 1\\ e^{-\frac{\left|C\right|-\left|C^{\prime }\right|}{T}} \end{array} & \begin{array}{l} \mathrm{for}\;\left|C^{\prime }\right|\geq \left|C\right|\\ \mathrm{otherwise} \end{array} \end{array}$$
where |C| is the size of the coverage of the area of interest of solution S, $\left|C^{\prime }\right|$ is the size of the coverage of transformed solution $S^{\prime }$, T is the current temperature and $p\left(S\to S^{\prime }\right)$ is the probability to replace solution S by solution $S^{\prime }$.

#### 3.3.2. Optimization of the Number of Waypoints

A simple algorithm to determine the minimum number of waypoints needed according to the optimization criterion in formula (4) is proposed in this section. The algorithm is shown in Figure 4 in pseudocode. $C_{min}$ is a minimal area of interest requested by the commander to be explored—see constraint (6). In point 3, the algorithm proposed to optimize positions of a certain number of waypoints from Section 3.3.1 is used (see Figure 3). This algorithm is used repeatedly until the termination condition is met and that is when the coverage exceeds the minimal requested coverage.

In point 2 of the algorithm, the first estimation of the minimum necessary number of waypoints is calculated according to formula (12). This estimation is based on the size of the area of interest and the maximum range covered from any waypoint. Then, locations of this number of waypoints are optimized using algorithm in Figure 3 (see point 3). If the coverage does not meet the condition for minimal coverage, a new estimation of the number is calculated according to formula (13) and the whole process is repeated. This new estimation is based on the previous number of waypoints and the portion of the area that is not yet covered.
(12)$$N=⎡\frac{C_{min}}{\pi \cdot R^{2}}⎤$$
(13)$$N=N+⎡(1-\frac{\left|C\right|}{C_{min}})\cdot N⎤$$
where N is the number of waypoints computed for the first time in formula (10) and updated in formula (11), R is the range of a sensor (see formula (2)), $C_{min}$ is the minimum requested size of the area of interest to be explored and |C| is the coverage of the area of interest from N waypoints (before updating).

## 4. Experiments and Results

This section is organized as follows. First (Section 4.1), a set of scenarios for experiments are defined and described. These scenarios have been proposed to verify the solution to the problem introduced in Section 3.3. All the scenarios have been configured to reflect typical parameters of real reconnaissance operations. Next (Section 4.2), the new approach is executed on the scenarios in order to show the improved efficiency achieved compared to the original CAR model. Then (Section 4.3), the new approach is compared to the algorithm proposed in the literature. Finally (Section 4.4), the results obtained using the new approach are evaluated in the situation which is not assumed by the CAR model but often possible—monitoring can be done not only from fixed waypoints but also continuously during the flight.

As already mentioned above, the experiments have been configured based on the typical reconnaissance operations. Thus, they have been placed into the real environment, which is based on the models as follows:Digital Elevation Model (DEM). This is a representation of the terrain surface in the form of a rectangular grid with the constant distance between squares (also known as a heightmap representing elevation). The DEM model used in these experiments has the distance between elevations 2.5 m. However, any DEM model would be possible to use, as in the current implementation of the algorithm, the linear interpolation is used to calculate the altitude in an arbitrary point of the terrain.Topographic Digital Data Model (TDDM). This is a database of topographic and other objects in the area; each object is represented by a polygon and parameters (e.g., object height). Buildings are selected as not-transparent objects that may occlude a certain portion of the area of interest to be explored.

### 4.1. Scenarios

Five different scenarios for reconnaissance operations were defined. Table 1 shows their basic parameters as follows—the size of the area of interest, the number of UASs available, the range of their sensors and their flight height, elevation difference between the lowest and the highest point inside the area of interest, the number of objects inside the area of interest and their average height. As can be seen, the flight height is constant in all scenarios, whereas the range changes; it means that different sensors are used in individual scenarios. The reason of this was to hold the flight height at a relatively low value to emphasize the occlusion effect. The individual scenarios can be described as in Table 2.

### 4.2. Experiments

#### 4.2.1. New Approach Compared to the Original CAR Model

In this section, the evaluation of the reconnaissance operations planned using the original CAR model in a complex environment was performed. The results were compared with the optimized deployment of waypoints using the algorithm proposed in Section 3.3.1; the same number of waypoints as determined by the original CAR model was used. The goal of this comparison was to verify the improved efficiency (i.e., the increasing coverage) which can be achieved using the new approach where the positions of waypoints are optimized. The parameters of the algorithm (see Figure 3) were set as follows: $T_{max}=100$, $T_{min}=10^{-5}$, α=0.9, k=500, r=50. The optimizations were executed 50 times and the best solution was recorded.

Table 3 shows the results. The total coverage of the area of interest, when the operation is planned by the original CAR model, exceeds 80% in all cases. The scenarios with medium density of obstacles (sc01, sc02, sc03) are above 90%, whereas scenarios with either very high density of obstacles (sc04) or very uneven terrain (sc05) are below 90%. When locations of the same number of waypoints are optimized, the coverage is improved in all cases. The biggest improvement was achieved in scenario sc05 where the coverage was improved by more than 8%. The average improvement is 4.0%. Table 3 also records the time for the reconnaissance operation, that is, the time needed to visit all waypoints by available UASs according to the optimization criterion in formula (7). As can be seen, this time did not differ much. In most cases (sc01, sc03, sc04, sc05), this time is slightly shortened when compared to the original deployment of waypoints.

The results shown in Table 3 illustrate the importance of the new approach proposed in this manuscript when the reconnaissance operation is to be performed in the area of operations with uneven terrain and/or obstacles. The original CAR model assumes that the terrain is perfectly flat with no obstacles; the optimization criterion of this model is to minimize the number of waypoints needed to cover the whole area of interest – but only in plane, that is, every point within the range of a sensor is assumed visible. The optimization criterion of the new model is also to minimize the number of waypoints to cover as large area as possible – but this time, the occlusion effect is taken into account and the visual line of sight to every point within the range of the sensor is determined.

#### 4.2.2. Optimization of Number of Waypoints

In this section, optimizations of the number of waypoints to achieve the requested minimum coverage $C_{min}$ were carried out using the improved CAR model (see the algorithm in Figure 4 in Section 3.3.2). The goal is to show the whole functionality of the improved CAR model which is to determine the minimum number of waypoints needed to cover at least the portion of the area of interest demanded by a commander. 

Table 4 shows the results for two values of the minimal coverage (in percent): $C_{min\;\%}=90\%$ and $C_{min\;\%}=98\%$. The optimizations were executed 50 times and the best solution was recorded. The algorithm worked correctly in all cases, the number of waypoints were determined to ensure the demanded coverage. The only exception was scenario sc04 and $C_{min\;\%}=98\%$. In this scenario, the area of interest was situated in a city centre with many tall buildings (up to 30 metres high) and narrow streets. Thus, a lot of small unexplored separated areas appeared and a large amount of waypoints was needed. For this reason, $C_{min\;\%}=97\%$ was chosen for this scenario. Even so, 285 waypoints were needed for this, which is 570 independent variables to be processed by the position optimization algorithm (limit in the current implementation of the algorithm is 600 variables). 

Figure 5 illustrates the results in scenario sc01. Figure 5a shows the area of interest (green colour) and objects (buildings). Figure 5b presents a real situation. The results recorded in Table 3 are depicted in Figure 5c (original CAR model) and 5d (optimized waypoints). The green area represents a covered (explored) area (darker green represents roofs of buildings); the red area is occluded (unexplored). The route of the UAS to visit all 16 waypoints is a violet line. In the same manner, the results from Table 4 are shown in Figure 5e ($C_{min\;\%}=90\%$) and Figure 5f ($C_{min\;\%}=98\%$).

### 4.3. Comparison

In this section, the improved CAR model is compared to the approach called the zig-zag algorithm introduced in Reference [38]. The goal is to illustrate the improvement in efficiency compared to the other frequently used approach when planning the reconnaissance or surveillance operations. The zig-zag algorithm is a simple approach based on the principles of the lawnmower pattern. The distance between adjacent rows (or columns) is set to $\sqrt{2R^{2}}$ where R is the range of sensors defined according to formula (2); the same distance is set between successive waypoints in a row (or column). In this way, the whole area is within the range of sensors.

Table 5 shows the results. As solutions for individual scenarios to be compared with the zig-zag algorithm, the solutions from Table 3 and Table 4, which differ least from the perspective of the number of waypoints, were selected. As can be seen, the bigger coverage was achieved from a lesser number of waypoints in all cases. In average, the solutions found by the improved CAR model proposed in this article use about 30% fewer waypoints but are able to cover 3.5% more of the area. Moreover, the time to conduct the reconnaissance operations is shorter in all cases (by more than 21%).

Figure 6 shows the situation in scenario sc01. The zig-zag algorithm monitors almost 93% of the area from 21 waypoints, whereas the improved CAR model is able to monitor more than 98% from only 18 waypoints. The reconnaissance operation can be conducted in 2:32 minutes when planned by the zig-zag algorithm and in 1:59 when planned by the CAR model.

### 4.4. Monitoring During a Flight

The CAR model assumes that monitoring the area can be done only when UASs are located at waypoints and not when moving between them. This section evaluates the coverage of the area of interest when monitoring is possible from any point during the flight of the UASs, that is, an infinite number of waypoints may be deployed along the routes of the UASs. This is done because the monitoring during flight is often a possibility in the real situations.

Table 6 shows the results based on the original CAR model (Table 3), the improved CAR model (Table 4) and the zig-zag algorithm (Table 4). The coverage increased significantly in all cases. The improved CAR model provides solutions that exceed 99% in all cases. Figure 7 shows the situation in case of scenario sc01.

## 5. Conclusions

In this article, the Cooperative Aerial Reconnaissance model, which was designed to plan reconnaissance operations conducted by a fleet of unmanned aerial systems, was improved to reflect a complex environment and uneven terrain. The contribution is not only scientific but also practical. From the scientific perspective, a new problem formulation of the CAR model was introduced and an optimization algorithm, based on the annealing simulation principles, was proposed. From the practical perspective, the improved CAR model was implemented in the Tactical Decision Support System. This system is designed to support tactical level commanders of the Czech Army with their decision-making processes.

To evaluate the effectiveness of the improved CAR model, 5 scenarios were set out. These scenarios were placed into a complex environment with uneven terrain and/or many obstacles and, at the same time, they can be characterized as typical reconnaissance operations. First, the evaluation of the original CAR model on the scenarios was performed (Table 3). The coverage of the area of interest was over 80% in all cases. Then, the positions of the waypoints were optimized using the proposed algorithm to maximize the coverage. The coverage increased in all cases; the smaller the original coverage, the bigger the improvement.

The most important experiments were carried out to verify the algorithm for finding the minimum number of waypoints required for the coverage of at least the portion of the area of interest demanded by the commander (Table 4). Two values of minimum coverage (90% and 98%) were tested. The algorithm managed to find the necessary number of waypoints and to optimize their locations so that the requested coverage might be achieved. In case of scenario sc04, the minimum requested coverage was lowered to 97% because too many waypoints were necessary. Their number would exceed 300, which is the limit in the current implementation of the algorithm.

The proposed principle was also compared to the zig-zag algorithm based on the lawnmower pattern (Table 5). The reconnaissance operations planned to use the improved CAR model need noticeably smaller number of waypoints to cover the bigger portion of the area of interest. Besides the total coverage, one of the most important criteria of the reconnaissance operation is the time needed for conducting the operation. From this point of view, the time of the operation planned by the improved CAR model is significantly shorter compared to the zig-zag algorithm. Finally, the CAR model was evaluated under the condition that monitoring can be done not only from waypoints but also during flights of UASs. The coverage increased significantly and exceeded 99% in most cases (Table 6).

In their future work, the authors will aim at converging the problems of reconnaissance to persistent surveillance, that is, monitoring the area of interest will be continuous. Similar principles introduced in this article can be used, for example, the optimization of waypoints locations to maximize the coverage of the area of interest. The collision avoidance of cooperating UASs will also be examined. The current model assumes that UASs have their own collision avoidance systems. Moreover, the routes of UASs should not cross if they were planned optimally.

## Figures and Tables

**Figure 1 sensors-19-03754-f001:**
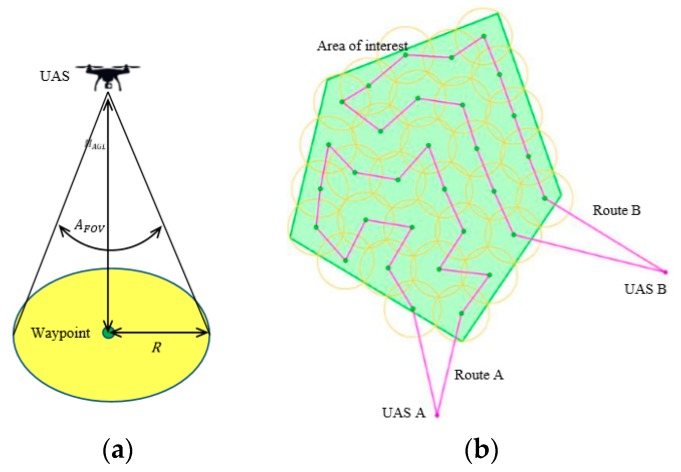
Cooperative Aerial Model: (**a**) monitoring from a waypoint; (**b**) example situation.

**Figure 2 sensors-19-03754-f002:**
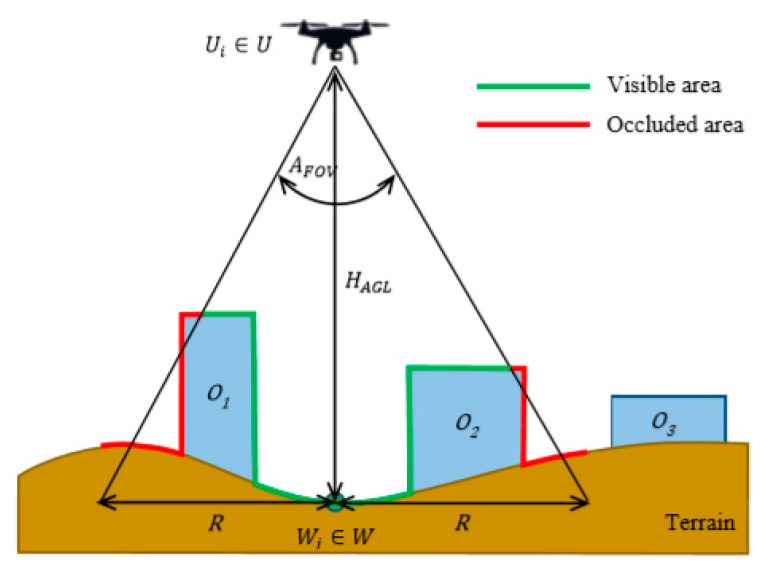
Coverage of the area by a sensor of an unmanned aerial system (UAS).

**Figure 3 sensors-19-03754-f003:**
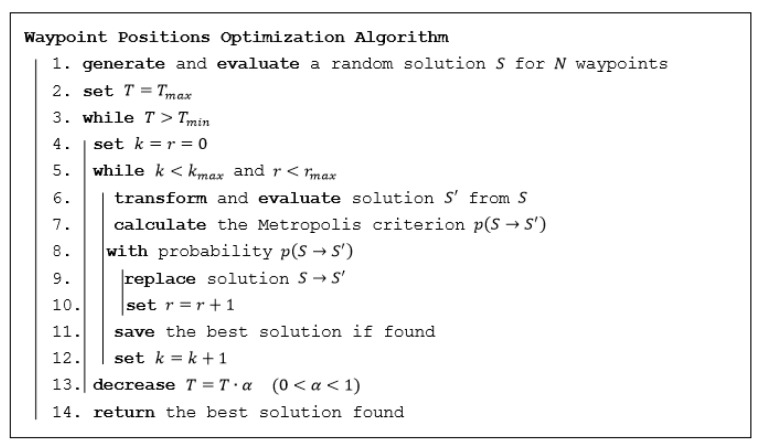
Simulated annealing used for the waypoint positions optimization problem.

**Figure 4 sensors-19-03754-f004:**
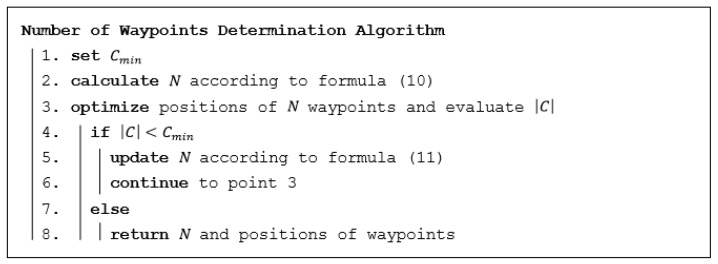
Algorithm for determining the number of waypoints.

**Figure 5 sensors-19-03754-f005:**
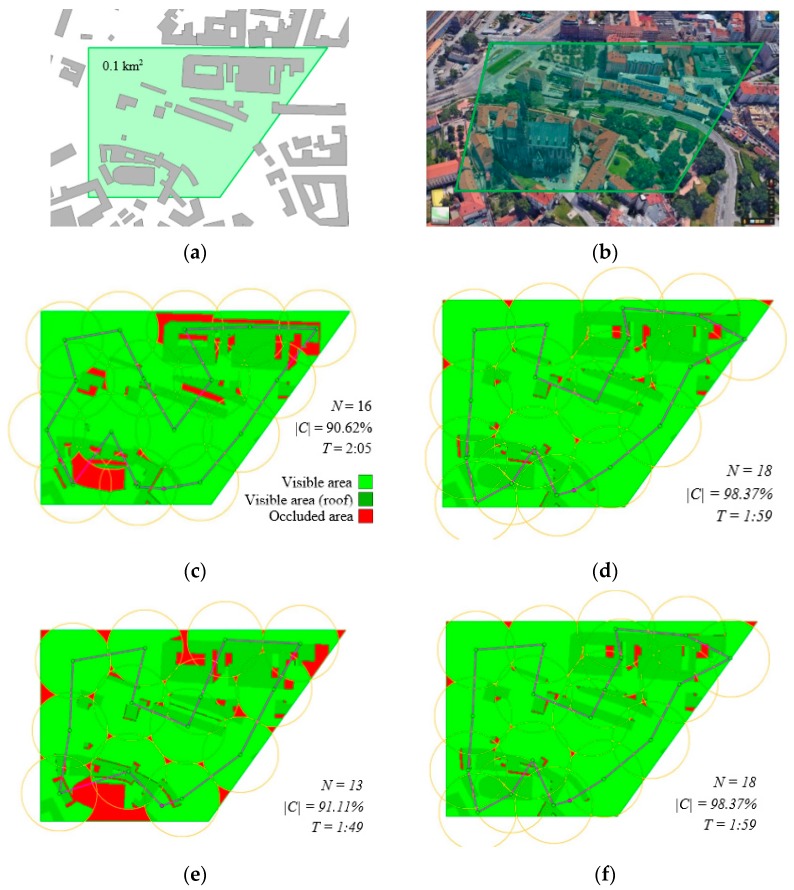
Scenario sc01: (**a**) area of interest and objects; (**b**) real environment; (**c**) original CAR model; (**d**) optimized waypoints; (**e**) solution by the improved CAR model for $C_{min\;\%}=90\%$; (**f**) solution by the improved CAR model for $C_{min\;\%}=98\%$.

**Figure 6 sensors-19-03754-f006:**
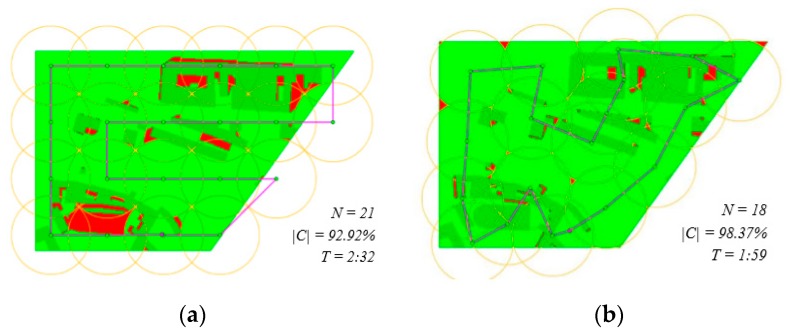
Scenario sc01: (**a**) solution by the zig-zag algorithm; (**b**) solution by the improved CAR model.

**Figure 7 sensors-19-03754-f007:**
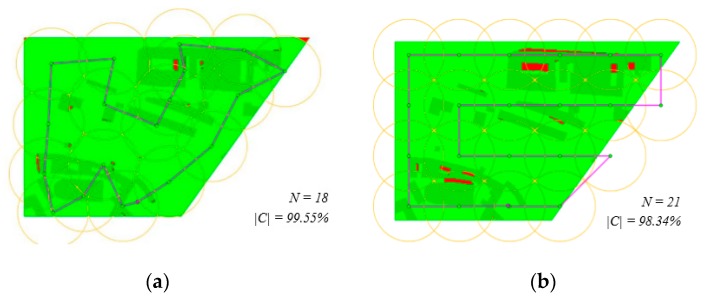
Scenario sc01 when monitoring is enabled during a flight: (**a**) solution by the improved CAR model; (**b**) solution by the zig-zag algorithm.

**Table 1 sensors-19-03754-t001:** Scenarios for experiments.

Scenario	Area of Interest		UASs		Elevation Difference	Objects	
		Count	Range	$H_{AGL}$		Count	Avg Height
sc01	0.1 km^2^	1	50 m	40 m	44 m	14	14.6 m
sc02	0.7 km^2^	2	100 m	40 m	79 m	266	6.4 m
sc03	2.8 km^2^	5	120 m	40 m	129 m	533	5.1 m
sc04	5.3 km^2^	3	130 m	40 m	47 m	936	10.6 m
sc05	6.8 km^2^	4	325 m	40 m	654 m	8	9.4 m

**Table 2 sensors-19-03754-t002:** Characteristics of scenarios.

Scenario	Characteristics
sc01	Very small area of interest of simple shape, medium density of tall buildings, uneven terrain (compared to the size of the area)
sc02	Small area of interest of simple shape, medium density of buildings of medium height, slightly uneven terrain
sc03	Medium-sized area of interest of very irregular shape, medium density of buildings of medium height, slightly uneven terrain
sc04	Large area of interest of irregular shape, very high density of tall buildings, narrow streets, flat terrain
sc05	Large area of interest of simple shape, very low density of buildings, very uneven terrain

**Table 3 sensors-19-03754-t003:** Comparison of the original Cooperative Aerial Reconnaissance (CAR) model with optimized waypoints.

Scenario	Number of Waypoints	Original CAR Model		Optimized Waypoints	
		Coverage	Op. Time	Coverage	Op. Time
sc01	16	90.62%	2:05	96.36%	1:58
sc02	31	95.49%	5:19	96.90%	5:38
sc03	94	97.78%	9:25	98.43%	9:20
sc04	150	82.19%	16:10	86.19%	15:44
sc05	30	89.65%	7:15	97.91%	6:51

**Table 4 sensors-19-03754-t004:** Optimization of the number of waypoints.

Scenario	$C_{min\;\%}=90\%$			$C_{min\;\%}=98\%$		
	Waypoints	Coverage	Op. Time	Waypoints	Coverage	Op. Time
sc01	13	91.11%	1:49	18	98.37%	1:59
sc02	23	90.14%	4:44	32	98.35%	5:21
sc03	67	90.13%	8:39	91	98.06%	9:09
sc04	175	90.29%	16:43	285	97.31% ^1^	20:36
sc05	23	90.23%	6:06	31	98.79%	7:08

^1^$C_{min\;\%}=97\%$.

**Table 5 sensors-19-03754-t005:** Comparison of the improved CAR model with the zig-zag algorithm.

Scenario	Zig-Zag Algorithm			Improved CAR Model		
	Waypoints	Coverage	Op. Time	Waypoints	Coverage	Op. Time
sc01	21	92.92%	2:32	18	98.37%	1:59
sc02	47	97.44%	6:15	32	98.35%	5:21
sc03	146	98.38%	12:39	94	98.43%	9:20
sc04	191	84.04%	20:12	175	90.29%	16:43
sc05	38	94.47%	7:44	31	98.79%	7:08

**Table 6 sensors-19-03754-t006:** Monitoring enabled during a flight.

Scenario	Original CAR Model		Improved CAR Model		Zig-Zag Algorithm	
	Waypoints	Coverage	Waypoints	Coverage	Waypoints	Coverage
sc01	16	97.90%	18	99.55%	21	98.34%
sc02	31	99.43%	32	99.59%	47	99.65%
sc03	94	99.69%	91	99.55%	146	99.68%
sc04	150	95.80%	285	99.05%	191	96.33%
sc05	30	98.07%	31	99.72%	38	99.04%

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
