# Peer review of "Cooperative Unmanned Aerial System Reconnaissance in a Complex Urban Environment and Uneven Terrain"

_sensors, 2019, doi:10.3390/s19173754_

Round 1

Reviewer 1 Report

In this manuscript, the cooperative UAV reconnaissance in a complex urban environment and uneven terrain is studied. It is well studied and does provide enough contributions. However, the theoretical analysis is not adequate. You should prove the performance of your proposed scheme through theoretical analysis. In addition, some of the latest works on this aspect are missed, e.g., "UAV-Assisted Emergency Networks in Disasters" and "UAV-Relaying-Assisted Secure Transmission With Caching".

Author Response

Dear reviewer, thank you very much for your helpful points.

The response to your comment is in the file.

All changes in the manuscript can be seen in the form of Word comments.

Reviewer 2 Report

The manuscript presents the results of proposes of using unmanned robotic systems in reconnaissance or survilance. Authors proposed and developed UAS reconnaissance model to increase its efficiency in a complex urban environment or uneven terrain.

The manuscript is written quite good, but the methodology and results must be extended. In Introduction, authors explained the proposed used technology. Introduction is written very well.

However, I have some important remarks to the manuscript:

Line 285: Please provide more details informations about DEM. Why was the 2.5 m resolution adopted? What is the source of this model's origin?

Line 292: How were the waypoints detected?

I have objections to the discussion section. The authors need to re-organize ,the results and discussion therein to better highlight to the reader what was done and what is relevant. The gain of the presented technique for the addressed application should be made more explicit in the form: What do the findings allow what was possible before. Authors should discuss the results and how they can be interpreted in perspective of previous studies and of the working hypotheses.

Conclusions are correct.

References contain a lot of autocytations ca. 25% This is not acceptable, please explain !

Finally, I think that the article would be interesting when the section 4 will be rewritten.

Author Response

Dear reviewer, thank you for your helpful comments.

The response to your comments is in the file.

All changes in the manuscript can be seen in the form of Word comments.
